# Characterization of the Complete Mitochondrial Genomes of Two Species with Preliminary Investigation on Phylogenetic Status of Zyginellini (Hemiptera: Cicadellidae: Typhlocybinae)

**DOI:** 10.3390/insects11100684

**Published:** 2020-10-10

**Authors:** Xian Zhou, Christopher H. Dietrich, Min Huang

**Affiliations:** 1Key Laboratory of Plant Protection Resources and Pest Management of Ministry of Education, Entomological Museum, Northwest A&F University, Yangling 712100, Shaanxi, China; zhouxian191@163.com; 2Illinois Natural History Survey, Prairie Research Institute, University of Illinois, 1816 S. Oak St., Champaign, IL 61820, USA; chdietri@illinois.edu

**Keywords:** mitochondrial genome, Typhlocybinae, Typhlocybini and Zyginellini, phylogenetic analysis

## Abstract

**Simple Summary:**

Typhlocybinae currently is the second largest membracoid subfamily distributed worldwide and includes many important agricultural pests. The monophyly of Typhlocybinae has been supported by several recent analyses, but the relationships of its included tribes remain largely unexplored, particularly that of the phylogenetic relationship of Zyginellini and Typhlocybini. This study presented the annotated complete mitochondrial genome sequences of two zyginelline species, *Limassolla* sp. and *Parazyginella tiani* Gao, Huang and Zhang, 2012, and a comparative analysis of mitochondrial genomes within the Typhlocybini and Zyginellini. Typhlocybinae mitogenomes are highly conservative in overall organization, as have been found in some other Cicadellidae. The only unusual feature was found in the secondary structure of tRNAs, with the acceptor stem of *trnR* comprising only 5 or 6 bp in some species. This unusual feature was reported for the first time in Typhlocybinae. Phylogenetic analyses showed that the monophyly of tribe Zyginellini was not supported and should be treated as a synonym of Typhlocybini. Nevertheless, a broader analysis with a much larger sample of taxa is needed to confirm the present results.

**Abstract:**

To explore the characteristics of mitogenomes and reveal phylogenetic relationships of the tribes of Zyginellini and Typhlocybini in Typhlocybinae, mitogenomes of two species of the Zyginellini, *Parazyginella tiani* and *Limassolla* sp., were sequenced. Mitogenomes of both species contain 13 protein-coding genes (PCGs), 22 transfer RNA genes (tRNAs), two ribosomal RNA genes (rRNAs) and a large non-coding region (A + T-rich region). These characteristics are similar to other Membracoidea mitogenomes. All PCGs initiate with the standard start codon of ATN and terminate with the complete stop codon of TAA/G or with an incomplete T codon. All tRNAs have the typical clover-leaf structure, except *trnS1* which has a reduced DHU arm and the acceptor stem of *trnR* is 5 or 6 bp in some species, an unusual feature here reported for the first time in Typhlocybinae. The A + T-rich region is highly variable in length and in numbers of tandem repeats present. Our analyses indicate that *nad6* and *atp6* exhibit higher evolutionary rates compared to other PCGs. Phylogenetic analyses by both maximum likelihood and Bayesian methods based on 13 protein-coding genes of 12 species of Typhlocybinae suggest that Zyginellini are paraphyletic with respect to Typhlocybini.

## 1. Introduction

Membracoidea is the largest hemipteroid superfamily, comprising nearly one-third of known hemipteran species, with approximately 24,000 valid species [1,2]. The leafhopper family Cicadellidae is among the 10 largest families of insects, and Typhlocybinae currently ranks as the second largest membracoid subfamily, with approximately 6000 described species [3,4,5]. The subfamily is distributed worldwide and includes many important agricultural pests such as the potato leafhopper (*Empoasca fabae* (Harris, 1841)), white apple leafhopper (*Zonocyba pomaria* (McAtee, 1926)), cotton leafhopper (*Amrasca biguttula* (Ishida, 1913)) and kaki leafhoppers (*Limassolla hebeiensis* Cai, Liang and Wang, 1992) [6,7,8].

The tribes of Typhlocybinae are most readily distinguished based on the wing venation. Although Oman et al. (1990) included ten valid tribes in Typhlocybinae [9], most recent eastern hemisphere workers recognize six tribes proposed by Dworakowska (1979): Alebrini, Dikraneurini, Empoascini, Erythroneurini, Typhlocybini and Zyginellini [10]. The monophyly of Typhlocybinae has been supported by several recent analyses [4,11,12,13], but the status and relationships of its included tribes remain largely unexplored, particularly that of the phylogenetic status of the tribe Zyginellini. Zyginellini and Typhlocybini were traditionally distinguished based on differences in hind wing venation [14] but more recently the former was considered to be a junior synonym of the latter [5]. The phylogenetic status and relationships of these two tribes have not yet been adequately tested by previous analyses of DNA sequence data and morphological characteristics [4,11,12]. Thus, additional data, such as complete mitogenomes, should be used to further investigate their phylogenetic relationships.

The insect mitogenome is typically a closed circular double-stranded DNA molecule, usually 15–18 kb in length and encoding 37 genes, including 13 protein-coding genes (PCG), 22 transfer RNA genes (tRNA), 2 ribosomal RNA genes (rRNA) and a control region (A + T-rich region) [15,16,17]. Because of maternal inheritance, absence of introns, high evolutionary rate, and rare recombination, the mitochondrial genome is considered ideal for phylogenetic and evolutionary analysis [18,19,20,21,22,23]. 

Currently, there are 14 complete or nearly complete mitogenomes of Typhlocybinae in GenBank [24,25,26,27,28,29,30,31,32,33,34,35]. To facilitate comparative studies and phylogenetic analyses, we sequenced the complete mitogenomes and provided functional annotations of two species in Zyginellini. Genomic structure, base composition, substitutional and evolutionary rates of six species of Typhlocybini and Zyginellini were comparatively analyzed. Combining mitogenome sequences of Typhlocybini and Zyginellini with previously available mitogenomes of other tribes of Typhlocybinae, i.e., Empoascini and Erythroneurini, we reconstructed phylogenetic relationships among major lineages of this subfamily to test the monophyly of Zyginellini and Typhlocybini. 

## 2. Materials and Methods 

### 2.1. Sample Preparation and DNA Extraction

Specimens of *Limassolla* sp. were collected at Hejiaping Village, Chongqing City, China and *Parazyginella tiani* Gao, Huang and Zhang, 2012 specimens were collected at Gutian Mountain, Quzhou City, Zhejiang Province, China. All specimens were preserved in 100% ethyl alcohol and stored at −20 °C in the Entomological Museum of the Northwest A&F University, Yangling, Shaanxi Province, China. Identification of adult specimens was based on the venation of the hind wing and male genitalia characteristics. Total DNA was extracted from muscle tissues of the thorax using the EasyPureR Genomic DNA Kit following the manufacturer’s instructions (TransGen, Beijing, China).

### 2.2. Sequence Analysis and Gene Annotation

Whole mitochondrial genome sequences of the 2 species, *Limassolla* sp. and *Parazyginella tiani*, were generated using an Illumina HiSeq platform with paired reads of 2 × 150 bp by the Biomarker Technologies Corporation (Beijing, China). First, the raw paired reads were quality-trimmed and assembled using Geneious 10.0.5 (Biomatters, Auckland, New Zealand) with default parameters [36] using the previously published mitochondrial genomes of *L. lingchuanensis* Chou and Zhang, 1985 and *P. luodianensis* Yuan and Song, 2019 as bait sequences for *Limassolla* sp. and *P. tiani*, respectively [25,26]. The genomes were annotated using Geneious 8.1.3 and *L. lingchuanensis* and *P*. *luodianensis* as references, respectively [25,26]. A total of 13 PCGs were identified as open reading frames based on the invertebrate mitochondrial genetic code; rRNA genes and control regions were identified by alignment with homologous genes from other Typhlocybinae species; tRNA genes were identified using the MITOS Web Server (http://mitos.bioinf.uni-leipzig.de/index.py) [37] and secondary structures were plotted with Adobe Illustrator CS5. Finally, a circular mitogenome map was drawn using CGView (http://stothard.afns.ualberta.ca/cgview_server/) [38]. Nucleotide composition, codon usage, composition skewness and relative synonymous codon usage (RSCU) were analyzed using PhyloSuite v 1.1.15 [39]. Tandem repeats in the A + T-rich region were predicted using the Tandem Repeats Finder program (http://tandem.bu.edu/trf/trf.html) [40]. The nucleotide diversity (Pi) and sliding window analysis (a sliding window of 200 bp and a step size of 20 bp) based on 13 aligned protein-coding genes (PCGs) were performed using DnaSP v 5.0. [41]. Kimura-2-parameter genetic distances were calculated using MEGA 7.0 [42]. Non-synonymous (dN) /synonymous (dS) mutation rate ratios among the 13 PCGs were calculated with DnaSP v 5.0. [41].

### 2.3. Phylogenetic Analysis 

Mitochondrial genomes of 12 species of Typhlocybinae were selected, including 2 species from Empoascini, 4 species from Erythroneurini, 2 species from Typhlocybini and 4 species from Zyginellini. Two species belonging to different cicadellid subfamilies were chosen as outgroups: *Scaphoideus varius* Vilbaste, 1968 (Deltocephalinae: Scaphoideini) and *Taharana fasciana* Li, 1991 (Coelidiinae: Coelidiini) (Table 1) [43,44]. All previously available mitochondrial genomes in this study were acquired from GenBank. Statistical phylogenetic analysis of mitogenomes was conducted using PhyloSuite v 1.1.15 [39]. The nucleotide sequences of all 13 PCGs and 2 rRNA genes were used in our analyses. PCGs were aligned using the MAFFT v 7.313 plugin in PhyloSuite v 1.1.15 [39]. Nucleotide sequences were aligned using the G-INS-i (accurate) strategy and codon alignment mode; the two rRNAs were aligned with the Q-INS-i strategy [45]. Gaps and ambiguous sites were removed using Gblocks v 0.91b [46], then the results were concatenated in PhyloSuite v 1.1.15 [39]. PartitionFinder2 was used to select the optimal partition schemes and substitution models [47]; the results are shown in Appendix A. We used the “greedy” algorithm with branch lengths estimated as “linked” and the Bayesian information criterion (BIC).

Alignments of individual genes were concatenated to generate 4 data sets: (1) the PCG12 matrix, including only the first and second codon positions of protein-coding genes; (2) the PCG12R matrix, including the first and second codon positions of protein-coding genes and the 2 rRNA genes; (3) the PCG123 matrix, including all 3 codon positions of protein-coding genes; (4) the PCG123R matrix, including all 3 codon positions of protein-coding genes and the 2 rRNA genes. 

Phylogenetic trees were constructed using the IQ-TREE web server [48] and MrBayes 3.2.6 [49] based on maximum likelihood (ML) and Bayesian inference (BI), respectively. The ML tree was constructed with the ML + rapid bootstrap (BS) algorithm with 1000 replicates. For the Bayesian analysis, the default settings were used with 5 × 10^6^ MCMC generations after reaching stationarity (average standard deviation of split frequencies <0.01), with estimated sample size >200, and potential scale reduction factor ≈1 [50].

## 3. Results and Discussion

### 3.1. Genome Organization and Base Composition

The complete mitogenome sequences of *P. tiani* and *Limassolla* sp. were 17,562 bp and 17,053 bp in length, respectively (Figure 1 and Figure 2). Known mitogenomes of Typhlocybinae range from 14,803 bp (*Illinigina* sp.) [27] to 16,497 bp (*P. luodianensis*) [25]. Length variation of Typhlocybinae mitogenomes is mainly due to variation in the size of the A + T-rich region. Each mitogenome includes the 37 standard mitochondrial genes; gene arrangement is the same as that of the hypothetical ancestral insect’s mitogenome in order and direction [51] (Figure 1 and Figure 2). Among the 37 mitochondrial genes, 23 genes (9 PCGs and 14 tRNAs) are transcribed from the majority strand (J-strand) and the remaining genes (four PCGs, eight tRNAs, and two rRNAs) are located on the minority strand (N-strand) (Appendix A). The base composition of *P. tiani* is A = 42.1%, T = 34.3%, C = 13.2%, G = 10.4% and that of *Limassolla* sp. is A = 41.7%, T = 33.3%, C = 14.0%, G = 11.0%. The A/T nucleotide composition is 76.4% in *P. tiani* and 75.0% in *Limassolla* sp., thus exhibiting a strong A/T bias. The two species in this study have a positive AT skew and a negative GC as in the other four previously sequenced species of Typhlocybini and Zyginellini (Table 2 and Table 3).

### 3.2. Protein-Coding Genes (PCGs)

The 13 PCGs of *P. tiani* and *Limassolla* sp. comprise a total of 10,938 bp and 10,926 bp, respectively. The A + T content of the third codon positions (86.7%, 83.5%) is much higher than that of the first (73.2%, 71.7%) and the second (68.8%, 69.1%) positions. The two sequenced species show negative AT skew (−0.126, −0.110) and negative GC skew (−0.041, −0.025) in PCGs (Table 2). The majority of PCGs in mitogenomes have the typical start codon ATN and end with the TAA stop codon or its incomplete form T-. Incomplete stop codons are common in insects and believed to be completed by posttranscriptional polyadenylation [52]. Gene *atp8* starts with TTG in *Typhlocyba* sp., *Bolanusoides shaanxiensis* Huang and Zhang, 2005, *L. lingchuanensis* and *P. luodianensis* [24,25,27]. Incomplete stop codon T- is present in *cox2* and *nad5* in *Typhlocyba* sp. [28]; *B*. *shaanxiensis* ends with T- in *cox2*, *cox3*, *nad5* and *nad4*; *L. lingchuanensis* ends with T- in *cox1*, *cox2* and *nad4* [24]; *P. luodianensis* ends with T- in *cox1*, *cox2*, *cox3* and *nad5* [25]; *Limassolla* sp. ends with T- in *cox1*, *cox2*, *cox3*, *nad4* and *nad6* and *P. tiani* ends with T- in *cox2*, *cox3*, *nad5* and *nad1*. Other genes end with a complete TAN codon (Table 4). Relative synonymous codon usage (RSCU) is summarized in Figure 3, indicating that the most frequently utilized amino acids are Leu, Ser, Ile, Phe and Met. Both newly sequenced species include all 62 available codons but the codons Pro (CCG) and Arg (CGC) are absent in *B. shaanxiensis*, Pro (CCG) and Arg (CGC/G) are absent in *P. luodianensis*. In the two new mitogenomes, as well as four other Typhlocybinae mitogenomes [24,25,28], the six most frequently used codons are all composed with A and T, which contribute to the high A + T bias of the entire mitogenomes.

### 3.3. Transfer and Ribosomal RNA Genes

Each of the sequenced mitogenomes includes 22 tRNA genes; the total length of tRNA of *P. tiani* and *Limassolla* sp. is 1422 and 1432 bp, respectively; 14 tRNAs are encoded on the J-strand and the remainder are encoded on the N-strand; and the tRNAs of the two species have a positive AT and GC skew (Table 2). Secondary structures of all tRNAs are the typical cloverleaf secondary structure, except for that of *trnS1*, which lacks the dihydrouridine (DHU) arm forming a simple loop, as commonly found in other insect mitogenomes [16,53]. A cloverleaf secondary structure is conservative: 7 bp in the acceptor stem, 5 bp in the anticodon arm, with length of the DHU and TΨC arms variable. While the acceptor stem of *trnR* was 6 bp in *Limassolla* sp., *L. lingchuanensis* and *Empoascanara sipra* Dworakowska, 1980 [24,26], this stem is 5 bp in *B. shaanxiens*, *Ghauriana sinensis* Qin and Zhang, 2011, *Illinigina* sp. and *Mitjaevia protuberanta* Song, Li and Xiong, 2011 [27,28,30]. This result has not been previously reported in Typhlocybinae but such characteristics may be due to mitochondrial genome variation among different individuals [54]. Base pair mismatch is commonly found in the tRNA secondary structure of Cicadellidae [55]. In the arm structures of tRNAs of the two new mitogenomes, we recognized a total of six types of unmatched base pairs (G–U, U–U, A–A, G–A, U–C, A–C). The total number of unmatched base pairs found was 35 and 40 in *P. tiani* and *Limassolla* sp., respectively (Figure 4 and Figure 5). The large and small ribosomal RNA genes are located between *trnL1* and *trnV*, and *trnV* and the A + T-rich region (Figure 1 and Figure 2). The lengths of rRNAs of *P. tiani* and *Limassolla* sp. are 1892 and 1976 bp and the AT content is 82.7% and 78.5%, respectively, with negative AT-skew and positive GC-skew (Table 2). 

### 3.4. Gene Overlaps and Intergenic Spacers

There are 11 and 15 intergenic spacers in *P.tiani* and *Limassolla* sp. ranging from 1 to 28 bp in size. The longest (28 bp) intergenic spacer is found in *Limassolla* sp., between the *nad5* and *trnH* genes. Gene overlaps range from 1 to 8 bp in the two mitogenomes, with the longest (8 bp) found between *trnW* and *trnC* genes in both species (Appendix A). tRNA genes have more gene overlaps, which may reflect fewer evolutionary constraints on such genes [56].

### 3.5. A + T-Rich Region

The A + T-rich region is the starting region of replication and the largest intergenic spacers are also believed to be involved in regulating transcription and replication of DNA in insects [57,58,59]. This region is located between *rrnS* and *trnM* in the two newly sequenced leafhoppers (Figure 1 and Figure 2). The length of this region ranges from 912 to 945 bp in Typhlocybini, and from 1416 to 3230 bp in Zyginellini (Table 3). One tandem repeat region is found in the control region of *P. tiani*, four Poly (T) are located in non-repeat regions. Three tandem repeat regions in the control region of *Limassolla* sp. include a Poly (T) and Poly (A) between the first and second tandem repeat region. The other four species have a single tandem repeat region. The length, nucleotide sequences and copy numbers of repeat units in the control region are highly variable among known Typhlocybinae mitochondrial genomes (Figure 6).

### 3.6. Nucleotide Diversity 

Sliding window analysis was implemented to study the nucleotide diversity of 13 PCGs among six mitogenomes exhibited in Figure 7A. Nucleotide diversity values range from 0.185 (*cox1*) to 0.333 (*atp8*). Among the genes, *atp8* (Pi = 0.333) has the highest variability, followed by *nad2* (Pi = 0.312), *nad6* (Pi = 0.265) and *nad4* (Pi = 0.238). In contrast, *cox1* (Pi = 0.185) and *nad4L* (Pi = 0.200) have comparatively low values and are the most conserved of the 13 PCGs. The nucleotide diversity is highly variable among the 13 PCGs. Pairwise genetic distances among these six mitogenomes produced results in Figure 7B. *atp8* (0.444), *nad2* (0.406) and *nad6* (0.382) evolve comparatively faster, while *cox1* (0.213), *nad4L* (0.234) and *nad1* (0.236) evolve comparatively slowly (Figure 7B). Average non-synonymous (Ka) and synonymous (Ks) substitution rates can be used to estimate the evolutionary rate [60]. The pairwise Ka/Ks analyses showed that the average Ka/Ks ratios (ω) of 13 PCGs are from 0.169 to 0.615 (0 < ω < 1) (Figure 7B), indicating that these genes are under purifying selection [61]. The genes *atp8* (0.615), *nad2* (0.499) and *nad5* (0.498) have comparatively high Ka/Ks ratios, while *cox1* (0.169), *cytb* (0.212) and *cox2* (0.281) have relatively low values (Figure 7B).

Nucleotide diversity analyses are crucial for designing species-specific markers useful for insects belonging to groups difficult to distinguish by morphology alone [62,63]. The mitochondrial gene *cox1* has long been used as a common barcode for identifying species and inferring the phylogenetic relationships in insects [64,65]. In Typhlocybinae, the *cox1* gene exhibits a relatively slow evolutionary rate compared to other PCGs, while *nad2* and *nad6* genes evolve more quickly. Thus, *nad2* and *nad6* may be more suitable as barcode genes for species identification in Typhlocybinae.

### 3.7. Phylogenetic Analyses

The best partitioning scheme and models for the different datasets as selected by PartitionFinder are listed in Appendix A. The phylogenetic topologies were highly consistent based on the four analyzed datasets (PCG123, PCG123R, PCG12 and PCG12R) with high nodal support values in BI and ML trees. The results (Figure 8) recovered Typhlocybinae as monophyletic. Empoascini, with *Empoasca* and *Ghauriana* sampled here, was resolved as monophyletic. Four included Erythroneurini, *Empoascanara sipra*, *Empoascanara dwalata* Dworakowska, 1971, *Illinigina* sp. and *Mitjaevia protuberanta*, forming a monophyletic clade sister to the clade comprising Typhlocybini and Zyginellini. All members of Typhlocybini and Zyginellini were grouped into a clade with high support (PP = 1, BS = 100), but the included Zyginellini formed a paraphyletic grade giving rise to Typhlocybini. Thus, the monophyly of tribe Zyginellini was not supported. 

The tribe Typhlocybini was proposed by Distant [14] and is currently divided into two informal groups according to the number of cross veins on the hind wing: the *Eupteryx* complex with three cross veins and another one with two cross veins, including the *Typhlocyba* complex, *Farynala* complex, and *Linnavuoriana* complex [66]. The tribe Zyginellini was proposed by Dworakowska based on the presence of one cross vein on the hind wing and with vein CuA apparently directly connected to MP [67], but its status as a separate tribe has been questioned [68]. Balme (2007) and Dietrich (2013) treated it as a synonym of Typhlocybini based on analysis of morphological characteristics and histone H3 and 16S rDNA sequences [4,5]. Recent phylogenetic analyses of Cidadellidae and Membracoidea [12,13] also grouped members of Zyginellini with Typhlocybini but these analyses did not include a large enough taxon sample to test the monophyly of the tribes. In our study, members of the *Typhlocyba* complex of Typhlocybini are derived from within Zyginellini, consistent with treatment of the two tribes as synonyms. Nevertheless, a broader analysis with a much larger sample of taxa is needed to confirm the present results.

## 4. Conclusions

This study presents the annotated complete mitochondrial genome sequences of *P. tinai* and *Limassolla* sp. and a comparative analysis of mitochondrial genomes within the Typhlocybini and Zyginellini. Typhlocybinae mitogenomes are highly conservative in overall organization, exhibiting the hypothetical ancestral gene order for insects and lacking any gene rearrangements, as have been found in some other Cicadellidae. The only unusual feature was found in the secondary structure of tRNAs, with the acceptor stem of *trnR* comprising only 5 or 6 bp in some species. This unusual feature is here reported for the first time in Typhlocybinae. Phylogenetic analyses support treating Zyginellini as a synonym of Typhlocybini, as proposed by other recent authors. However, a much larger taxon with mitogenomes is still needed to reconstruct a better phylogeny tree to solve the tribal relationships within Typhlocybinae.

## Figures and Tables

**Figure 1 insects-11-00684-f001:**
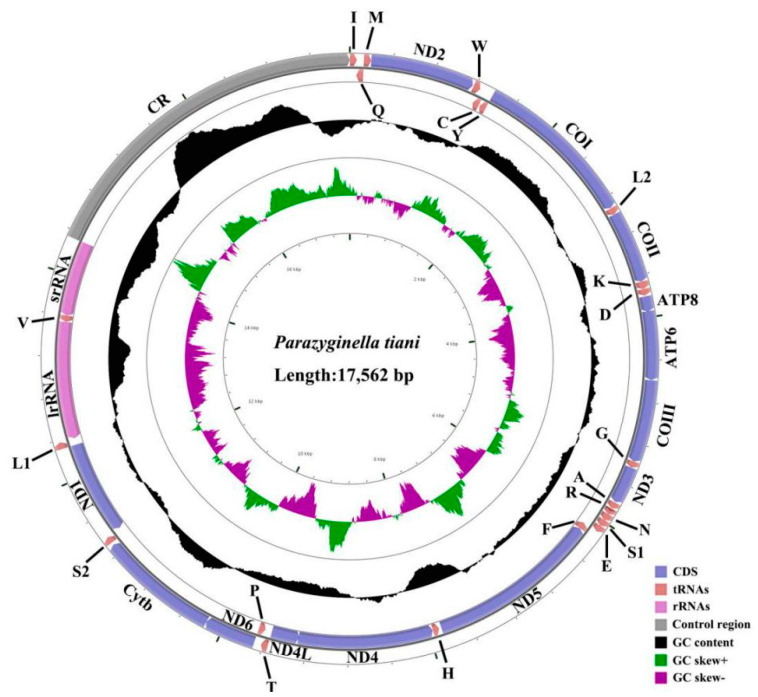
Organization of the complete mitogenome of *P. tiani.*

**Figure 2 insects-11-00684-f002:**
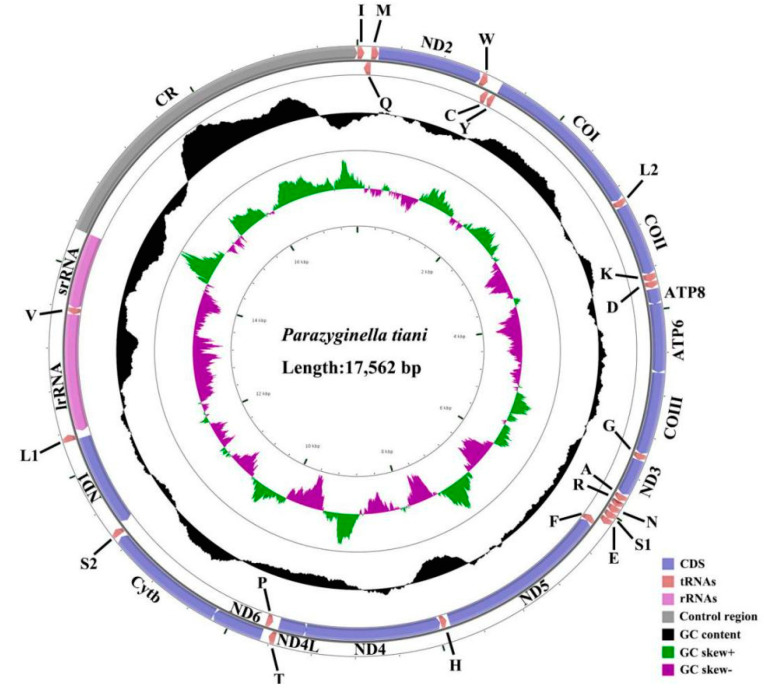
Organization of the complete mitogenome of *Limassolla* sp.

**Figure 3 insects-11-00684-f003:**
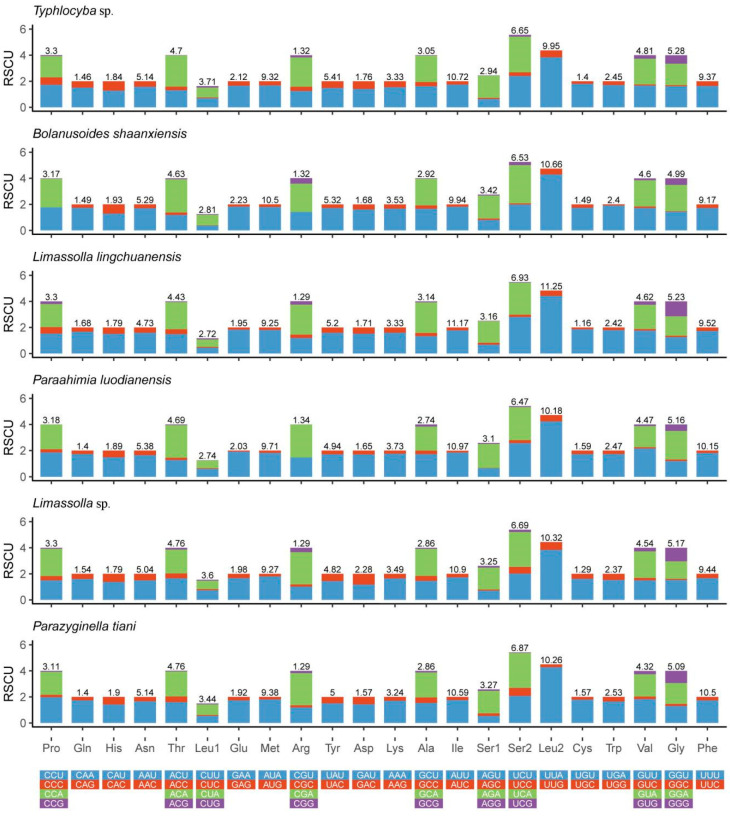
Relative synonymous codon usage (RSCU) in the mitogenomes of 6 species.

**Figure 4 insects-11-00684-f004:**
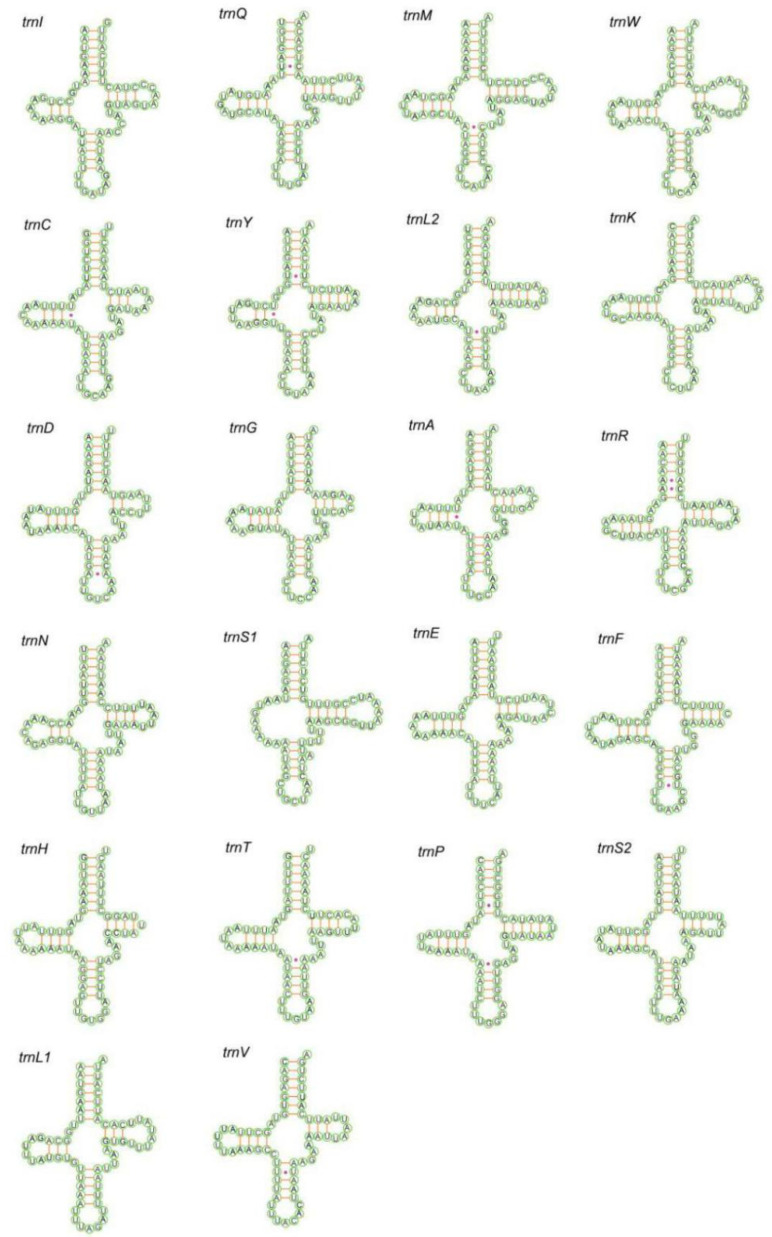
Predicted secondary cloverleaf structure for the tRNAs of *P. tiani.*

**Figure 5 insects-11-00684-f005:**
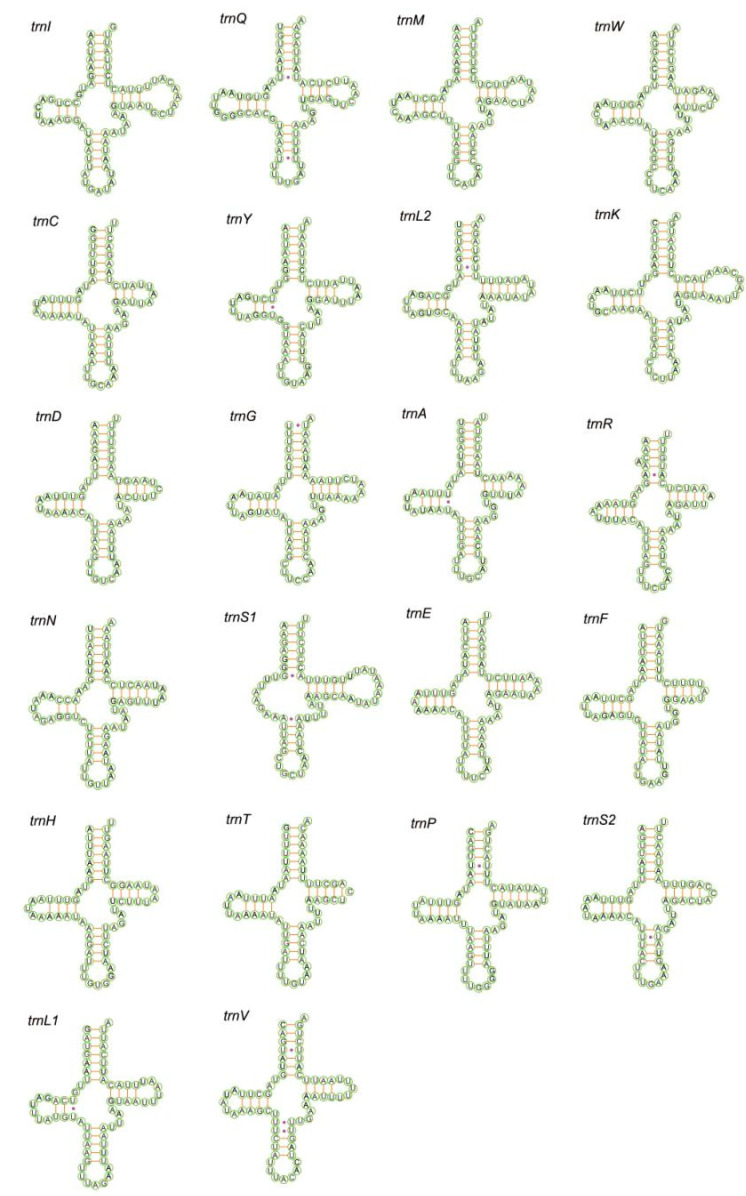
Predicted secondary cloverleaf structure for the tRNAs of *Limassolla* sp.

**Figure 6 insects-11-00684-f006:**
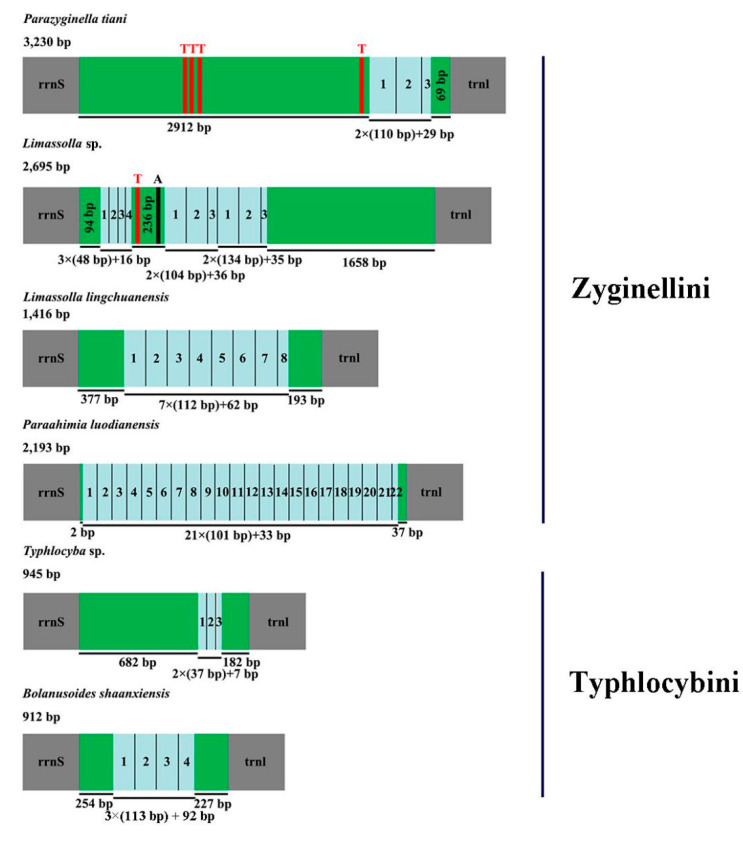
Organization of the control regions in the mitochondrial genomes of Typhlocybini and Zyginellini. The boxes colored blue indicate the tandem repeats. Non-repeat regions are shown by green boxes. The red and black blocks represent the structures of poly (T) and poly (A), respectively.

**Figure 7 insects-11-00684-f007:**
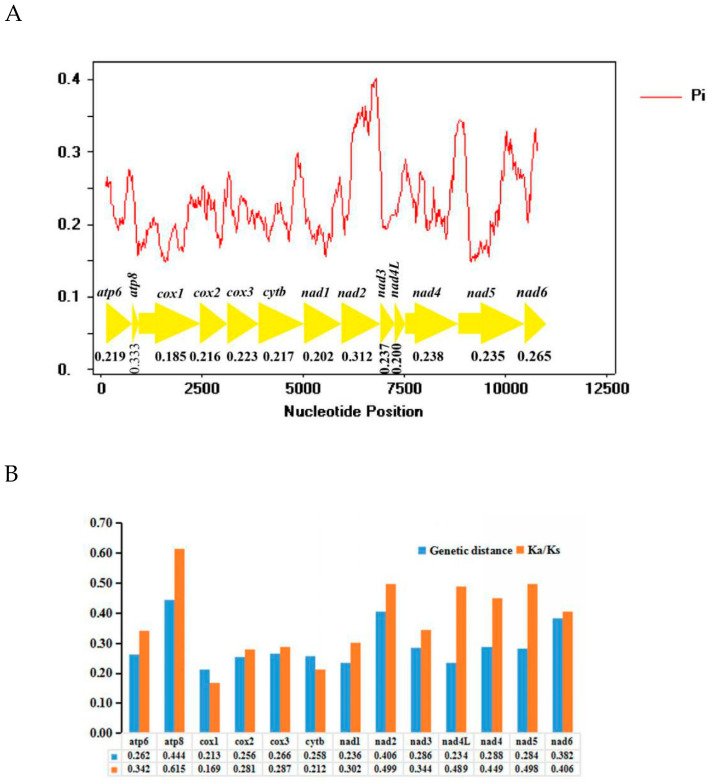
(**A**) Sliding window analyses of 13 protein coding genes among 6 species’ mitogenomes. The red line shows the value of nucleotide diversity (Pi) in a sliding window analysis (a sliding window of 200 bp with the step size of 20 bp); the Pi value of each gene is shown under the gene name. (**B**) Genetic distances and the ratio of non-synonymous (Ka) to synonymous (Ks) substitution rates of 13 protein-coding genes among 6 species of Typhlocybini. The average value for each PCG is shown under the gene name.

**Figure 8 insects-11-00684-f008:**
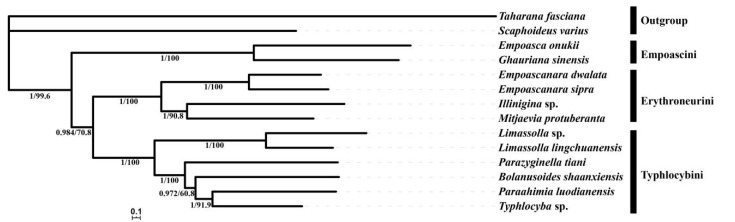
Phylogenetic tree produced from ML and BI analysis based on 13 protein-coding genes. Numbers on branches are Bayesian posterior probabilities (PP) and bootstrap values (BS) (The Typhlocybini including former Zyginellini).

**Table 1 insects-11-00684-t001:** Summary of mitogenomic sequence information used in the present study.

Subfamily	Tribe	Species	Accession Number	Reference
Coelidiinae	Coelidiini	*Taharana fasciana*	NC_036015	[44]
Deltocephalinae	Scaphoideini	*Scaphoideus varius*	KY817245	[43]
Typhlocybinae	Empoascini	*Ghauriana sinensis*	MN699874	[28]
		*Empoasca onukii*	NC_037210	[32]
	Erythroneurini	*Mitjaevia protuberanta*	NC_047465	[30]
		*Illinigina* sp.	KY039129	[27]
		*Empoascanara dwalata*	MT350235	[33]
		*Empoascanara sipra*	MN604278	[26]
	Typhlocybini	*Typhlocyba* sp.	KY039138	[27]
		*Bolanusoides shaanxiensis*	MN661136	Unpublished
	Zyginellini	*Limassolla lingchuanensis*	MN605256	[24]
		*Paraahimia luodianensis*	NC_047464	[25]
		*Limassolla* sp.	MT683892	This study
		*Parazyginella tiani*	MT683891	This study

**Table 2 insects-11-00684-t002:** Nucleotide composition and skewness of mitogenomes of *P. tiani* and *Limassolla* sp.

Regions	Size (bp)	T(U)	C	A	G	AT (%)	GC (%)	AT Skew	GC Skew
Full genome	17,562/17,053	34.3/33.3	13.2/14.0	42.1/41.7	10.4/11.0	76.4/75.0	23.6/25.0	0.103/0.112	−0.117/−0.119
PCGs	10,938/10,926	42.9/41.5	12.4/12.9	33.3/33.3	11.4/12.3	76.2/74.8	23.8/25.2	−0.126/−0.110	−0.041/−0.025
1st codon position	3646/3642	36.9/35.1	13.4/11.5	36.3/36.6	15.7/16.8	73.2/71.7	26.8/28.3	−0.008/0.022	0.172/0.187
2nd codon position	3646/3642	48.4/48.0	14.4/17.6	20.4/21.1	13.7/13.3	68.8/69.1	31.3/30.9	−0.407/−0.389	−0.123/−0.137
3rd codon position	3646/3642	43.4/41.4	15.4/9.7	43.3/42.1	4.8/6.7	86.7/83.5	13.3/16.4	−0.002/0.008	−0.278/−0.182
tRNAs	1422/1432	36.6/40.2	16.4/8.4	40.6/39.7	12.2/11.8	77.2/79.9	22.7/20.2	0.052/0.006	0.077/0.170
rRNAs	1892/1976	47.1/46.1	17.4/8.5	35.6/32.4	10.8/13.0	82.7/78.5	17.4/21.5	−0.139/−0.173	0.244/0.209
A + T rich-region	3230/2695	38.6/35.0	18.4/14.7	34.0/36.3	13.4/14.0	72.6/71.3	27.4/28.7	−0.063/0.019	−0.020/−0.026

**Table 3 insects-11-00684-t003:** Nucleotide composition of the Typhlocybini and Zyginellini mitochondrial genomes.

Species	Whole Genome		AT Skew	GC Skew	PCGs		tRNAs		rRNAs		A + T-Rich Region	
	Size (bp)	AT (%)			Size (bp)	AT (%)	Size (bp)	AT (%)	Size (bp)	AT (%)	Size (bp)	AT (%)
*B.*	15,274	78.9	0.158	−0.135	10,917	77.2	1469	79.7	1963	83.6	912	88.7
*T.*	15,223	77.1	0.138	−0.138	10,950	75.2	1436	77.9	1866	82.3	945	85.9
*L1.*	15,716	78.8	0.096	−0.098	10,932	76.3	1426	80.1	1911	83.1	1416	91.4
*P1.*	16,497	80.0	0.155	−0.162	10,962	77.7	1427	79.0	1898	83.7	2193	88.4
*L2.*	17,053	75.0	0.112	−0.119	10,926	74.8	1432	79.9	1976	78.5	2695	71.3
*P2.*	17,562	76.4	0.103	−0.117	10,938	76.2	1422	77.2	1892	82.7	3230	72.6

*B. shaanxiensis* (*B.*); *Typhlocyba* sp. (*T.*); *L. lingchuanensis* (*L1.*); *P. luodianensis* (*P1.*); *Limassolla* sp. (*L2.*); *P. tiani* (*P2.*).

**Table 4 insects-11-00684-t004:** Start and stop codons of the Typhlocybini and Zyginellini mitochondrial genomes. *B. shaanxiensis* (*B*.); *Typhlocyba* sp. (*T*.); *L. lingchuanensis* (*L1*.); *P. luodianensis* (*P1*.); *Limassolla* sp. (*L2*.); *P. tiani* (*P2*.).

Gene	Start Codon/Stop Codon					
	*T.*	*B.*	*L1.*	*P1.*	*L2.*	*P2.*
*nad2*	ATA/TAA	ATT/TAA	ATT/TAG	ATT/TAA	ATA/TAG	ATA/TAA
*cox1*	ATG/TAA	ATG/TAA	ATG/T	ATG/T	ATG/T	ATG/TAA
*cox2*	ATT/T	ATT/T	ATT/T	ATG/T	ATC/T	ATA/T
*atp8*	TTG/TAA	TTG/TAA	TTG/TAA	TTG/TAA	ATC/TAA	ATA/TAA
*atp6*	ATG/TAA	ATA/TAA	ATG/TAA	ATG/TAA	ATG/TAA	ATA/TAA
*cox3*	ATG/TAA	ATG/T	ATG/TAA	ATG/T	ATG/T	ATG/T
*nad3*	ATT/TAA	ATT/TAA	ATA/TAA	ATA/TAA	ATA/TAA	ATA/TAA
*nad5*	ATG/T	TTG/T	ATA/TAA	ATG/T	ATT/TAA	ATT/T
*nad4*	ATG/TAA	ATA/T	ATG/T	ATG/TAA	ATG/T	ATG/TAG
*nad4L*	ATG/TAA	ATG/TAA	ATG/TAA	ATG/TAA	ATG/TAA	ATG/TAA
*nad6*	ATT/TAA	ATT/TAA	ATA/TAA	ATT/TAA	ATA/T	ATT/TAA
*cytb*	ATG/TAG	ATT/TAA	ATG/TAA	ATG/TAA	ATG/TAA	ATG/TAA
*nad1*	ATT/TAA	ATT/TAA	ATT/TAA	ATT/TAA	ATT/TAA	ATA/T

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
