# Peer review of "Characterization of the Complete Mitochondrial Genomes of Two Species with Preliminary Investigation on Phylogenetic Status of Zyginellini (Hemiptera: Cicadellidae: Typhlocybinae)"

_insects, 2020, doi:10.3390/insects11100684_

Round 1

Reviewer 1 Report

Just one correction- in the Simple Summary, line 2, replace "distributing" with "distributed".

Author Response

Point 1: Just one correction- in the Simple Summary, line 2, replace "distributing" with "distributed".

Response: Line 14: Accepted and replaced "distributing" with "distributed".

Thank you very much for your comments.

Reviewer 2 Report

All my previous comments were successfully answered.

Author Response

Thank you very much for your comments.

This manuscript is a resubmission of an earlier submission. The following is a list of the peer review reports and author responses from that submission.

Round 1

Reviewer 1 Report

This study used mitochondrial genome sequences to resolve the phylogenetic relationships in the leafhopper subfamily Typhlocybinae. It utilises 14 already-published mitogenomes whilst contributing well-annotated mitogenomes for 2 more species. The authors find evidence for the monophyly of Typhlocybini but the Zyginellini were revealed to be paraphyletic. Ka/Ks analyses of the mitochondrial genes by the authors could lead to the identification of new, more specific, barcodes for the Typhlocybinae which may assist future phylogenies with more taxa.

Broad comments:

I would appreciate more background to the Typhlocybinae in the introduction so non-experts in the can better understand the rationale for resolving their taxonomy using mitogenomes.

The authors conclude by saying a much larger taxon size is needed to reconstruct tribal relationships in the Typhlocybinae. Would they expect a larger taxon size to support their findings and do they believe this should be done with mitogenomes or by desigining barcodes based on the genes they suggest?

Is it necessary to present quite so many large tables in the article? Some of them feel like they would be better placed in the Supplementary Materials The same could be said about Figures 3-6.

Specific comments:

Line 36: I’m not overly keen on the use of casual phrases like “…and so on” in scientific literature. I would prefer it if the authors could specify individual species rather than assuming specific knowledge on behalf of their reader.

Line 116: What was the longest mitogenome of Typhlocybinae before the two produced in this study? The inclusion of mitogenomes from this study when discussing the lengths of known mitogenomes feels a little misleading.       

Figure 4/5: I can’t read the names of the specific tRNAs

Figure 6: It would be nice to have the tribe highlighted next to the species names

Figure 8: Does the figure reduce Zyginellini and Typhlocybini into one tribe? This should be noted in the legend if it does as it is otherwise not clear where Zyginellini is in the phylogeny.

Line 75: repetition of “genetic”

Line 174: “This regions…” should be “This region…” or “These regions…”?

Reviewer 2 Report

The presented paper “Characterization of the Complete Mitochondrial Genomes of Two Species with Preliminary Investigation on Phylogenetic Status of Zyginellini (Hemiptera: Cicadellidae: Typhlocybinae)” by Xian Zhou and colleagues, describes and characterize two new mitogenomes of species which are involved in an economically important subfamily. Their research improves our knowledge on the phylogenetic origin of Typhlocybinae subfamily. This question is well presented in the introduction. Although is extremely short, it is clear and makes the point for the study. Methods are appropriate and clearly presented. Results are properly discussed and easy to read.

However, I just want to point out some minor observations 

Line 36: maybe it would be nice to the reader if the authors could provide the species names.

Line 51: “mitogenomes and and provide functional annotations of 2 species in Zyginellini” eliminate repeated “and”.

Line 98: I think that must be said that the partitions where set for each gene. As could be set by codon position also. Although it is showed on the table, I think that is ok to clarified that.

Table S1 has swapped title names.

Reviewer 3 Report

This study publishes the mitochondrial genomes of two species of Typhlocybinae and explores the classification of tribes Typhlocybini and Zyginellini, the latter of which has recently been considered a syn. of the former. The study is sound and the details of the genomic study are very good. 

One important issue concerns the species name for which one of the mt genomes is published.  The species name has not yet been published, and before the name is used here, it must be published.  If not, then some other temporary reference for this species must be used.  I will also note here that new species of Limassolla was just recently published (Oh et al., 2020) and should be checked to ensure it is not the species referred to in this work.

Please see marked .pdf with more notes and corrections to the text, figures, and tables.
